# The relationship between wasting and stunting in Cambodian children: Secondary analysis of longitudinal data of children below 24 months of age followed up until the age of 59 months

Mueni Mutunga[1]*, Alexandra Rutishauser-Perera[2], Arnaud Laillou[3], Sophonneary Prak[4], Jacques Berger[5], Frank T. Wieringa[5], Paluku Bahwere[6]

1 United Nations Children's Fund (UNICEF) East Asia Pacific Regional Office, Bangkok, Thailand, 2 Action Against Hunger UK, London, United Kingdom, 3 United Nations Children's Fund (UNICEF), Addis Ababa, Ethiopia, 4 National Nutrition Program, Maternal and Child Health Center, Phnom Penh, Cambodia, 5 Institut de Recherche pour le Développement, Montpellier, France, 6 Centre de Recherche en Epidémiologie, Biostatistique et Recherche Clinique, Ecole de santé publique, Université Libre de Bruxelles, Brussels, Belgium

* mmutunga@unicef.org

## Abstract

The interrelationship between wasting and stunting has been poorly investigated. We assessed the association between two indicators of linear growth, height-for-age Z-score (HAZ) change and occurrence of accelerated linear growth, and selected indicators of wasting and wasting reversal in 5,172 Cambodian children aged less than 24 months at enrolment in the 'MyHealth' study. The specific objectives were to evaluate the relationship between temporal changes in wasting and 1) change in HAZ and 2) episodes of accelerated linear growth. At enrolment, the stunting and wasting prevalence were 22.2 (21.0;23.3) % and 9.1 (8.1;10.1) %, respectively, and reached 41.4 (39.3;43.6) %, and 12.4 (11.5;13.3) % respectively, two years later. Between 14–19% of stunted children were also wasted throughout the whole study period. For each centimetre increase in Mid-Upper Arm Circumference (MUAC) from the previous assessment, the HAZ increased by 0.162 (0.150; 0.174) Z-score. We also observed a delayed positive association between the weight for height Z score (WHZ) unit increase and HAZ change of +0.10 to +0.22 units consistent with a positive relationship between linear growth and an increase in WHZ occurring with a lag of approximately three months. A similar positive correlation was observed for the occurrence of an episode of accelerated linear growth. These results show that interventions to prevent and treat wasting can contribute to stunting reduction and call for integrated wasting and stunting programming.

**Data Availability Statement:** All relevant data are within the manuscript and its Supporting Information files.

**Funding:** The authors received no specific funding for this work.

**Competing interests:** The authors have declared that no competing interests exist.

# Introduction

Wasting, defined by a low weight-for-height Z-score (WHZ < -2z) or mid-upper arm circumference (MUAC < 12.5cm), affected 47 million children globally in 2019. Stunting, defined by height-for-age Z-score (HAZ < -2z), affected 144 million children [1]. Many countries are not on track to meet the Sustainable Development Goal (SDG) 2.2 targets to reduce the wasting prevalence to below 5% and the number of stunted children by 40% by 2025 compared to 2012 [1–6]. These two forms of undernutrition often coexist in the affected countries and communities as they share similar risk factors [7–10]. The 2014 Cambodia Demographic Health Survey (CDHS) found a prevalence of wasting and stunting among children below five years of age of 10% and 32.5%, respectively [11]. These figures indicate that both conditions are of public health importance in Cambodia [6, 7]. Similar findings have been documented in other countries in the region [6, 9, 12, 13].

Despite the coexistence of wasting and stunting, the relationship between the two conditions has not attracted sufficient interest from the research community, program implementers or policymakers. Previous research focuses on the similarities of the determinants [7], associations of multiple nutritional deficits and mortality risk [7, 12, 14–16] and prevalence of concurrent wasting and stunting [12, 13, 16, 17].

We are aware of only one published peer-reviewed paper that checked whether wasting was a risk factor for stunting occurrence and vice-versa [17]. This study retrospectively analyzed data from well-run Gambian rural growth monitoring clinics collected from 1976 to 2016. It concluded that more attention on the interrelationship between wasting and stunting is required and proposed that both forms of undernutrition be addressed jointly [17].

Currently, wasting and stunting are addressed through separate programmes with limited integration [18–22]. Due to the separation between wasting and stunting programming, the effect of weight replenishment on linear growth has not been examined. This is despite hospital data since the 80's suggesting a link between the level of weight replenishment and linear catch-up growth in children recovering from wasting, with catch up growth only commencing when wasting has been addressed [23, 24]. Unfortunately, the separation of wasting and stunting programming is unlikely to change unless more evidence is produced. In this study, we analyzed longitudinal data from three provinces in Cambodia. The specific objectives of this study were to evaluate the relationship between temporal changes in wasting and 1) change in height-for-age and 2) episodes of accelerated linear growth. We aimed to answer two key research questions: 1) Are changes in indicators of wasting (WHZ and MUAC) associated with a change in HAZ velocity over time?, and, 2)Are the presence or absence of the same wasting parameters associated with the occurrence of accelerated linear catch-up growth in subsequent follow-up visits?

# Methods

## Study design and participants

This paper is based on a secondary analysis data from the "MyHealth" open cohort intervention study, which took place between February 2016 and August 2018 in six districts from three provinces in Cambodia; Russei Kaev in Phnom Penh, Chitr Borie and Krong Kratie in Kratie province, and Ou Chum, Krong Ban Lung, and Bar Kaev in Ratanakiri province. The provinces represent different population groups from Cambodia. The district in Phnom Penh concentrates a poor suburban population being from a diversity of ethnicities and religions. Kratie, a province crossed by the Mekong river, is inhabited by a population consisting of ethnic minorities that is highly reliant on agriculture. Ratanakiri province is concentrating ethnic

groups practicing subsistence agriculture and food collection from wild sources. Detailed methods of the study have been previously published [25–28]. The study enrolled all women from villages sampled from the six districts who were pregnant or lactating at any of the data collection rounds and surveyed them prospectively for over three years. In the same communities, children below three years were enrolled in the MyHealth study at the first round of data collection (baseline). For this paper, we have analyzed data of children who were less than 24 months at enrolment as they are the focus of most interventions aiming at tackling stunting. Hence, we excluded children aged ≥24 months at recruitment and those children who were surveyed only once. Subsequently, infants born from participating pregnant women were enrolled at the nearest data collection round after their birth. After recruitment, children were surveyed every 3 to 4 months during the first two years of the study and at six months intervals thereafter. Children exited the study when they reached five years or at the end of the study depending on whichever occurred first.

The measurements were taken by 8 teams of surveyors who received trained on anthropometric measurements for up to a maximum of 5 days for each round of data collection. Weight, length or height, and mid-upper arm circumference (MUAC) were measured in duplicates for each child, and the mean values were further used. The measurement tools were calibrated after each fifth measurement. A spot check team monitored the data collection teams' measurements techniques and calibration of the tools.

## Source of data

Anonymized data were extracted from the database of the "MyHealth" study described above, including; study identifier (child, household and mother identities); administrative information (province, district and date of the survey for the different data collection points); household characteristics (head of household level of education, mother level of education, number of people in the household, number of children below five years in the household, number of children below 15 years, type of toilet facility, type of source used to get drinking water); child socio-demographic characteristics (sex, birth date, and age at the different data collection points); and child nutrition parameters (weight, height, MUAC, weight-for-age Z-score (WAZ), HAZ and WHZ for the different data collection points).

## Analysis

**Variable transformation and definitions.** "MyHealth" study was a longitudinal open cohort design whereby children entered and exited the cohort at any of the follow up visits. Data were reorganized to ensure that the recruitment visit became 'Visit 0' for all children, regardless of the data collection round at which the children were recruited into the study. Subsequent visits were labelled as follow up visits.

The standard cut-off of -2 Z-score of the multiple countries' growth standards reference was used to define the nutrition status of children. Height-for-age Z-score (HAZ) <-2 were classified as stunting and weight-for-height Z-score (WHZ) <-2 as wasting [29].

Household and child characteristics were compared between children included and excluded from the analysis to evaluate potential selection bias.

Based on the age categories used by the Demographic Health Surveys (DHS), We defined the following three age categories: 0–5 months, 6 to 11 months and 12 to 23 months. We assumed that the level of education of both the mothers and heads of households (HHH) did not vary during the study period. Information from the follow-up visits (FV) were used to fill any missing variables data collected at recruitment. The following education level groups were defined: 0 to 6 years of education, 7 to 9 years and ≥10 years for the mothers, and 0 years, 1 to

6 years, 7 to 9 years and ≥10 years head of household (HHH). Household (HH) size was defined in four groups: <4 people, 4 to 5 people, 6 to 10 people and ≥11 people.

We created the variables 'HAZ increment' (HAZd), 'WHZ increment' (WHZd) and 'MUAC increment' (MUACd) by calculating the absolute difference between two consecutive visits of HAZ, WHZ and MUAC, respectively. We also created the binary variables 'Ever had acute malnutrition and 'Ever had an episode of accelerated linear growth' to distinguish those who experienced the condition at any data collection point. Accelerated linear growth was defined using the cut-off proposed in the literature of +0.67 HAZ increase between two data collection points [30–32]. Based on this cut-off, a child who had an HAZd ≥ 0.67 was considered as having experienced an accelerated linear growth [30].

We used the previous follow-up period to indicate the interval between the two last data collection points (visits) and the current follow-up period to indicate the previous and current FV interval period.

**Outcomes of interest and sample size.**   The primary outcomes of interest were HAZ change and accelerated linear growth as defined by HAZd≥0.67. Secondary outcomes of interest were the prevalence of stunting, the prevalence of wasting, and the prevalence of concurrence of wasting and stunting (WaSt).

There was no specific calculation of sample size for all the outcomes of interest for this paper. The sample size was calculated based on the main objective of the original study. We assumed that a sample of 1200 children below two years of age per region was necessary for the study to have sufficient power to demonstrate at least a 6% reduction in stunting prevalence over a three-year period. Children aged ≥24 months at recruitment and children who were surveyed only once were excluded from our sample.

**Data management and analysis.**   The data extracted from the "MyHealth" study database was exported into excel files. These files were then converted to STATA format and merged using STATA software version 14.1. Data cleaning, restructuring, missing values analysis, new variables creation and data analysis were performed using the same STAT 14.1 statistical package. The restructuring of the data consisted of arranging the data in order to have all recruitment (first assessment) data positioned at visit (V0) for children first surveyed at Follow up Visits one (FV1) to six (FV6). The analysis of missing values was done to determine the extent of missing information for each variable and identify time-varying variables for which data were not collected systematically in all the data collection time points. This analysis allowed the exclusion of variables not routinely collected and those associated with a considerable reduction in sample size in multivariate analyses. No missing values handling approach was used in our analysis as we assumed that for all the variables retained, the data were missing completely at random and opted for the complete cases analysis approach [33, 34].

We used standard statistics for quantitative analyses. Continuous variables were described using means and their 95% confidence interval (CI) and compared using the Student t-test. Categorical variables were described using proportion and their 95%CI and compared using Chi-square tests. To test for the linearity of the trend over time, we calculated a chi-square statistic for the trend using the STATA ptrend command [35]. We determined all the effect sizes, including prevalence, mean difference, univariate and adjusted odds ratio (OR) using multilevel linear or logistic mixed-effect modelling as appropriate to account for clustering at the regional level. The covariates included in the full initial multilevel linear or logistic regression mixed-effects models were selected based on their relevance according to the literature and availability in the received dataset]. We did not use stepwise methods to select the covariates in the multivariate analysis to determine the final models but excluded the non-significant covariates manually, as proposed by Greenland [36]. For fitting the mixed-effects models, we implemented the panel data analysis and used the STATA standard command Mixed and melogit as

appropriate [37, 38]. Based on the authors' observation from several countries on catch up growth after a hunger season or during recovery from an episode of wasting, linear catch-up growth occurs around three months after that of weight, starting only when the weight deficit is almost completely replenished. Therefore, we also conducted repeated cross-sectional modelling that allowed the introduction of one follow up the period lag in our models assessing the association between wasting and stunting [23, 39–41]. This corresponded to a time lag of approximately 4-months and meant that the covariates wasting and WHZ at FV1 were included in the model assessing the association between stunting and wasting at FV2. Such time lag was not introduced for MUAC as we did not have similar evidence. For this alternative modelling approach, we also used the standard stata command mixed and melogit. The results of this second modelling approach are mostly presented in S1 and S2 Tables.

## Ethical considerations

Ethical approval for the "MyHealth" study was granted by the Cambodia National Ethics Committee for Health Research (NECHR), National Institute of Public Health, Ministry of Health, Cambodia (number 117/NECHR). All candidates were informed in their local languages about study objectives and procedures, the voluntary aspects of participation, the possibility to withdraw consent at any given points, the research team's obligation to preserve the participants' privacy, and the use of data for scientific data publications. All participants provided written informed consent at baseline before their inclusion in the original MyHealth study. Consent was obtained from adults primary caregivers (mostly mothers) for participating children. Community health volunteers witnessed the entire participation approval process. The authorization to use the anonymized "MyHealth" study data for secondary data analysis and publication was granted by the relevant authority from the Cambodian Ministry of Health. The need for a new ethical approval for the current study was waived by the NECHR of the Ministry of Health.

## Results

### Study participants

Of the 5172 children included in this analysis, 70.3% (n = 3635) were recruited at baseline, 5.8% (n = 304) at follow up visit 1 (FV1), 5.8% (n = 298) at FV2, 2.9% (n = 151) at FV3, 4.1% (n = 210) at FV4 and 11.1% (574) at FV5. The median number of survey rounds for which the children were surveyed was 5 (interquartile range:3–6); Only 13.3% (n = 686) were surveyed in all the seven rounds.

Table 1 presents households, demographic and nutritional and child characteristics of children included in this analysis, based on data collected during recruitment into the study. About 40% of children were recruited in the Kratie region and about 30% in the other two areas. Most mothers (64.5%) and heads of households (60.1%) had between zero and six years of formal education. The average household size was about six people per household. Over three quarters of the households had an improved source of drinking water, while improved sanitation was available for less than 60% of the households. The great majority of children used mosquito nets. Male and female were equally represented, and the vast majority were less than 12 months of age at recruitment. One out of 10 children had a birth weight of less than 2.5kg. The median duration of exclusive breastfeeding was two months only, and the maximum duration reported was four months. A significant proportion of the children were underweight, stunted and wasted at recruitment into the cohort (Table 1).

Overall, 5.8% (299/5172) of children had received treatment for Severe Acute Malnutrition (SAM) during the follow-up period as reported by their caregivers. The proportion of children

**Table 1. Households and children characteristics.**

| Characteristics | n/N | % | Average |
|---|---|---|---|
| **Households** | | | |
| Region | | | |
| Phnom Penh | 1448/5172 | 28.0 | |
| Kratie | 2051/5172 | 39.7 | |
| Ratanakiri | 1673/5172 | 32.3 | |
| Mother formal education | | | |
| 0–6 Years | 3270/5069 | 64.5 | |
| 7–9 Years | 1227/5069 | 24.2 | |
| ≥10 Years | 572/5069 | 11.3 | |
| HHH formal education | | | |
| 0 Year | 908/3878 | 23.4 | |
| 1–6 Years | 1423/3878 | 36.7 | |
| 7–9 Years | 950/3878 | 24.5 | |
| ≥10 Years | 597/3878 | 15.4 | |
| Household size (n people) | | | |
| Mean number of people (SD[1]) | | | 5.8 (2.8) |
| <4 | 698/4219 | 16.5 | |
| 4–5 | 1695/4219 | 40.2 | |
| 6–10 | 1568/4219 | 37.2 | |
| ≥11 | 258/4219 | 6.1 | |
| Drinking water source | | | |
| Improved | 4070/5172 | 78.7 | |
| Sanitation | | | |
| Improved | 2911/5158 | 56.4 | |
| Mosquito net use | | | |
| Yes | 2256/3108 | 72.6 | |
| Sometimes | 632/3108 | 20.3 | |
| No | 220/3108 | 7.1 | |
| **Children** | | | |
| Child gender | | | |
| Girls | 2599/5172 | 50.2 | |
| Boys | 2573/5172 | 49.8 | |
| Birth weight(kg) | | | |
| Mean (SD) | | | 3.0(0.5) |
| <2.5 kg | 326/3204 | 10.2 | |
| ≥2.5kg | 2878/3204 | 89.8 | |
| Age at recruitment | | | |
| Mean (SD) | | | 9.7(6.8) |
| 0–5 months kg | 2004/5172 | 38.8 | |
| 6–11 months | 1284/5172 | 24.8 | |
| 12–23 months | 1884/5172 | 36.4 | |
| ≥ 24 months | 0/5172 | | |
| EBF[2] duration (months) | | | |
| Median (IQR[3]) | 5129 | | 3(2–3) |
| Weight at recruitment | | | |
| Mean (SD) | 5109 | | 7.2(2.0) |
| Height at recruitment | | | |

(*Continued*)

**Table 1.** (Continued)

| Characteristics | n/N | % | Average |
|---|---|---|---|
| Mean (SD) | 5093 | | 67.7(9.3) |
| Weight-for-age Z-score | | | |
| Mean (SD) | 5102 | | -1.1(1.1) |
| % Z-score <-2 | 1039/5102 | 20.4 | |
| Height-for-age Z-score | | | |
| Mean (SD) | 5075 | | -1.0(1.3) |
| % Z-score <-2 | 1026/5075 | 20.2 | |
| Weight-for-height Z-score | | | |
| Mean (SD) | 5074 | | -0.7(1.1) |
| % <-2 Z-score | 5074 | | |
| MUAC[4] (cm) | | | |
| Mean (SD) | 5051 | | 13.4(1.3) |
| <11.5 | 367/5051 | 7.3 | |
| 11.5–12.4 | 736/5051 | 14.6 | |
| ≥12.5 | 3948/5051 | 78.1 | |

[1]Wasted = weight-for-length/height Z-score<-2 (2006 WHO reference curves);

[2]Stunted = Length/height-for-age Z-score<-2 (2006 WHO reference curves);

[3]WaSt = Concurrently wasted and stunted;

[4]Mean = Standard deviations;

[5]CI = confidence interval;

[6]Visit 0 = recruitment into the cohort visit;

[7]FV = follow up visit;

[8]All age group cohorts combined;

[9]Both sex combined.

for whom treatment of SAM between 2 follow-ups was reported was 1.8% (81/4489), 1.0% (33/3359), 2.0 (61/3105), 5.4% (107/1977) and 4.6% (69/1492) for FV1, FV2, FV3, FV5 and FV6, respectively. This also shows that some children had been on SAM treatment for more than one period.

## Trends in the prevalence of wasting, stunting, and wasting and stunting concurrence

The regional weighted prevalence of wasting, stunting and wasting and stunting concurrence (WaSt) are presented in Table 2. Prevalence of wasting was always close or above 10%, with the highest prevalence observed at recruitment and the lowest at FV4. No linear trend was observed from recruitment to FV6 for wasting prevalence. For stunting, the prevalence was just over 20% at recruitment and linearly increased over time (p for linear trend <0.001), with the highest prevalence being observed at FV5. For WaSt, the prevalence was below 5% at recruitment into the cohort but increased linearly over subsequent follow-up visits (p for linear trend<0.001), crossing the 5% cut-off at FV 3.

Boys had a higher prevalence of wasting than girls at recruitment and at FV1 to FV4 but not at FV5 and FV6. The differences were statistically significant at recruitment [95% CI = 2.8 (1.0; 4.6) %; p = 0.002] and FV2 [95% CI = 4.0 (2.1; 6.0) %; p<0.001]. Boys also had a higher prevalence of stunting than girls from recruitment to FV4, but the difference was statistically significant.

**Table 2. Prevalence of wasting, stunting and concurrence of wasting and stunting at the different rounds of data collection by gender and age.**

| Data collection round | n | Age (m) [4] | Wasted[1] % (95%CI[5]) | Stunted[2] % (95%CI) | WaSt[3] % (95%CI) |
|---|---|---|---|---|---|
| All age groups & sex | | | | | |
| Visit 0[6] | 5174 | 9.6 (6.7) | 12.4 (11.5–13.3) | 22.2 (21.0–23.3) | 4.3 (3.8–4.9) |
| FV[7]1 | 4447 | 14.2 (7.0) | 10.0 (9.1–10.9) | 27.8 (26.5–29.2) | 4.5 (3.9–5.1) |
| FV2 | 3357 | 17.4 (6.7) | 9.6 (8.6–10.6) | 32.2 (30.6–33.8) | 4.7 (4.0–5.4) |
| FV3 | 2955 | 21.8 (6.6) | 11.2 (10.0–12.3) | 35.6 (33.8–37.3) | 6.4 (5.5–7.3) |
| FV4 | 3087 | 26.8 (6.6) | 9.1 (8.1–10.1) | 36.7 (35.0–38.4) | 5.3 (4.5–6.1) |
| FV5 | 2041 | 38.0 (7.0) | 10.4 (9.1–11.8) | 41.4 (39.3–43.6) | 5.9 (4.8–6.9) |
| FV6 | 1496 | 47.5 (6.7) | 11.4 (9.7–13.0) | 36.5 (34.1–39.0) | 5.4 (4.2–6.5) |
| Boys[8] | | | | | |
| Visit 0 | 2573 | 9.7 (6.8) | 13.4 (10.6–16.3) | 22.6 (16.1–29.1+) | 4.6 (3.4–5.8) |
| FV1 | 2195 | 14.3 (7.0) | 10.5 (9.2–11.8) | 30.7 (28.8–32.6) | 5.1 (4.1–6.0) |
| FV2 | 1670 | 17.5 (6.7) | 11.5 (10.0–13.1) | 33.5 (31.3–35.8) | 6.0 (4.8–7.1) |
| FV3 | 1456 | 21.8 (6.7) | 11.6 (10.0–13.3) | 37.2 (35.0–39.7) | 7.1 (5.8–8.4) |
| FV4 | 1520 | 26.9 (6.7) | 9.3 (7.8–10.7) | 37.7 (35.3–40.2) | 5.4 (4.3–6.6) |
| FV5 | 1010 | 37.9 (7.0) | 10.1 (8.3–12.0) | 40.9 (37.8–43.9) | 5.7 (4.3–7.1) |
| FV6 | 733 | 47.5 (6.7) | 10.2 (8.0–12.4) | 35.9 (32.4–39.4) | 5.2 (3.6–6.8) |
| Girls[8] | | | | | |
| Visit 0 | 2599 | 9.5(6.7) | 10.6 (7.8–13.5) | 17.6 (11.1–24.1) | 3.3 (2.1–4.5) |
| FV1 | 2252 | 14.1 (7.0) | 9.6 (8.4–10.8) | 25.1 (23.3–26.9) | 4.0 (3.1–4.8) |
| FV2 | 1687 | 17.4 (6.8) | 7.7 (6.4–9.0) | 30.9 (28.7–33.2) | 3.4 (2.6–4.3) |
| FV3 | 1499 | 21.8 (6.7) | 10.8 (9.2–12.4) | 34.0 (31.6–36.4) | 5.7 (4.5–6.9) |
| FV4 | 1567 | 26.7 (6.5) | 8.9 (7.5–10.3) | 35.7 (33.3–38.1) | 5.2 (4.1–6.3) |
| FV5 | 1031 | 38.2 (7.1) | 10.8 (8.8–12.7) | 42.0 (38.9–45.0) | 6.0 (4.6–7.5) |
| FV6 | 763 | 47.4(6.8) | 12.5 (10.1–14.9 | 37.2 (33.7–40.6) | 5.6 (3.9–7.2) |
| Age 0–5 months[9] | | | | | |
| Visit 0 | 2004 | 2.9 (1.6) | 7.4 (6.3–8.6) | 13.0 (11.5–14.5) | 0.7 (0.3–1.1) |
| FV1 | 1742 | 8.0 (3.6) | 7.2 (6.0–8.5) | 18.0 (16.2–19.8) | 1.9 (1.2–2.5) |
| FV2 | 1386 | 12.0 (4.5) | 9.2 (7.7–10.7) | 26.4 (24.1–28.7) | 3.6 (2.6–4.6) |
| FV3 | 1149 | 16.3 (4.5) | 13.5 (11.5–15.4) | 33.0 (30.2–35.7) | 7.8 (6.2–9.3) |
| FV4 | 1061 | 21.7 (5.7) | 10.2 (8.4–12.0) | 37.0 (34.1–39.9) | 6.7 (5.1–8.2) |
| FV5 | 641 | 31.3 (3.6) | 13.5 (10.8–16.1) | 44.7 (40.8–48.6) | 8.4 (6.2–10.5) |
| FV6 | 417 | 39.5 (2.2) | 14.9 (11.4–18.3) | 41.5 (36.8–46.3) | 8.1 (5.4–10.7) |
| Age 6–11 months[9] | | | | | |
| Visit 0 | 1284 | 8.8 (1.7) | 11.4 (9.5–13.2) | 20.8 (18.6–23.1) | 3.0 (2.1–4.0) |
| FV1 | 1123 | 13.5 (3.6) | 12.8 (10.9–14.8) | 29.3 (26.6–31.9) | 5.6 (4.2–6.9) |
| FV2 | 784 | 16.5 (3.2) | 11.6 (9.3–13.8) | 32.3 (29.0–35.6) | 5 6(4.0–7.2) |
| FV3 | 712 | 20.2 (2.7) | 11.3 (8.9–13.6) | 36.4 (32.9–40.0) | 7.2 (5.3–9.1) |
| FV4 | 764 | 24.0 (2.2) | 8.4 (6.4–10.4) | 35.9 (32.4–39.3) | 5.4 (3.8–7.1) |
| FV5 | 508 | 35.8 (3.7) | 8.8 (6.3–11.3) | 41.8 (37.5–46.1) | 5.1 (3.0–7.2) |
| FV6 | 402 | 45.2 (2.1) | 10.5 (7.5–13.5) | 37.1 (32.4–41.9) | 5.3 (3.1–7.5) |
| Age 12–23 months[9] | | | | | |
| Visit 0 | 1884 | 17.4 (3.3) | 18.5 (16.7–20.3) | 33.0 (30.8–35.1) | 9.1 (7.8–10.4) |
| FV1 | 1582 | 21.5 (4.2) | 11.2 (9.6–12.8) | 37.8 (35.4–40.2) | 6,7 (5.4–7.9) |
| FV2 | 1187 | 24.4 (3.5) | 8.8 (7.2–10.4) | 39.2 (36.4–42.0) | 5.4 (4.1–6.7) |
| FV3 | 1094 | 28.6 (3.5) | 8.7 (7.1–10.4) | 37.5 (34.8–40.6) | 4.4 (3.2–5.7) |
| FV4 | 1262 | 32.5 (3.7) | 8.5 (6.9–10.0) | 36.9 (34.2–39.6) | 4.0 (2.9–5.1) |

*(Continued)*

**Table 2.** (Continued)

| Data collection round | n | Age (m) [4] | Wasted[1] % (95%CI[5]) | Stunted[2] % (95%CI) | WaSt[3] % (95%CI) |
|---|---|---|---|---|---|
| FV5 | 892 | 44.3 (4.6) | 9.1 (7.2–11.0) | 38.8 (35.5–42.0) | 4.4 (3.0–5.7) |
| FV6 | 677 | 53.7 (3.6) | 9.6 (7.4–11.9) | 33.1 (29.6–36.7) | 3.7 (2.3–5.2) |

[1]Wasted = weight-for-length/height Z-score<-2 (2006 WHO reference curves);

[2]Stunted = Length/height-for-age Z-score<-2 (2006 WHO reference curves);

[3]WaSt = Concurrently wasted and stunted;

[4]Mean = Standard deviation,

[5]CI = confidence Interval;

[6]Visit 0 = recruitment into the cohort visit;

[7]FV = follow up visit;

[8]All age group cohorts combined;

[9]Both sex combined.

The highest prevalence of wasting was observed at 39.5 (SD = 2.2) months of age for the 0 to 5 months age group cohort (FV6), at 13.5 (SD = 3.6) months for the age group cohort 6 to 11 months (FV1) and 17.4 (SD = 3.3) months for the age group cohort 11 to 23 months at recruitment (V0). For stunting, the highest prevalence was observed at the mean age of 31.3 (3.6) months for the age group cohort 0 to 5 months (FV5), 35.8 (3.7) months for the age group cohort 6 to 11 months (FV5) and 24.4 (3.5) months for the age group cohort 12 to 23 months (FV2).

The analysis of trends showed that the prevalence of wasting increased linearly for age group cohort 0 to 5 months [slope(SE) = 1.3 (0.1) %; p<0.001] and age group cohort 6 to 11 months [slope(SE) = 0.3 (0.1) %; p = 0.027], while it decreased linearly for the age group cohort 12 to 23 months [slope(SE) = -1.4 (0.1) %; p<0.001]. For stunting prevalence, there was a significant linear trend for the age group cohort 0 to 5 months [slope(SE) = 5.7 (0.2) %; p<0.001] and 6 to 11 months age group cohort [slope(SE) = 3.1 (0.2) %; p<0.001], but not for the 12 to 23 months age group cohort [slope(SE) = 0.2 (0.2) %; p = 0.163]. The proportion (95%CI) of stunted children who were WaSt did not show any linear trend.

### Association between wasting parameters (presence of wasting, WHZ change, MUAC change) and HAZ change

At FV1, non-wasted children (n = 3829) had a smaller decrease in HAZ than wasted children (n = 813) at recruitment: [-0.23 (0.75) for the non-wasted versus -0.44 (0.90) for the wasted; [95% CI = 0.21 (0.14; 0.28); P<0.001]. A similar relationship was observed at FV2 [-0.15 (0.57) for the non-wasted(n = 2667) at FV1 versus -0.23 (0.63) for the wasted (n = 260) at FV1 [95% CI = 0.08 (0.01, 0.15); p = 0.031]; FV3 [-0.10 (0.52) for the non-wasted (n = 2199) versus -0.15 (0.49) for the wasted(n = 228) at FV2 [95% CI = 0.05 (-0.01; 0.12); p = 0.190]; and FV4 [-0.06 (0.46) for the non-wasted (n = 2143) versus -0.11 (0.51) for the wasted (n = 224) at FV3 [95% CI = 0.05 (-0.01; 0.12); p = 0.111. The observed difference was not significant for FV3 and FV4. For FV5 the relationship observed was inverse to that observed at FV1 to FV4 but not significant [-0.07 (0.57) for the non-wasted (n = 1611) versus -0.05 (0.40) for the wasted (n = 162); at FV4 [95% CI = -0.02 (-0.11; 0.07); p = 0.697]. At FV6, in both non-wasted and wasted at FV5, there was an increase rather than a decrease in HAZ suggesting a catch up in linear growth in both groups, with the magnitude being significantly higher in the non-wasted

children: [0.13 (0.41) for the non-wasted (n = 963) versus 0.04 (0.33) for the wasted (n = 110) at FV5 [95% CI = 0.09 (0.01; 0.16); p = 0.033].

Fig 1 below shows the change in HAZ over time for children categorized into four groups: no anthropometric deficit (good), only wasted, only stunted, and wasted and stunted. At all the FVs, the most significant decrease in HAZ was observed for children who were only wasted at the previous visit (Fig 1). In contrast, some catch up in linear growth (increase in HAZ) was observed for children with no anthropometric deficit, only stunted and wasted and stunted, with the largest catch up always observed for children only stunted (Fig 1). The difference in change of HAZ between those with no anthropometric deficit and children only wasted varied between -0.5 and -0.30 Z-score, with the smallest difference observed at FV3 and the largest difference observed at FV1. For the comparison between those with no anthropometric deficit and those who were only stunted at the previous follow up visit (FV), the observed HAZ differences varied between +0.15 Z-score at FV6 and _+0.51 Z-score at FV1. The differences between children with no anthropometric deficit and those who had both wasting and stunting at the previous FV varied between 0.0 Z-score at FV6 and +0.31 Z-score observed at FV1. The observed differences were all significant (p<0.05) except for the comparison of those without the anthropometric deficit and wasted at FV3 and FV6, and for those without a deficit and the wasted and stunted children at FV6.

Further, multilevel multivariate longitudinal analysis adjusting for socio-demographic and economic status at baseline (age group, sex, region of residence, mother level of education, type of drinking water source, type of toilet facility), anthropometric characteristics at baseline *MUAC group, HAZ group and WHZ group)MUAC <12.5 cm or not, and the presence or absence of stunting at the previous visits showed that an increase in MUAC between two follow-up visits was associated with an increase in HAZ, while the inverse was observed for WHZ

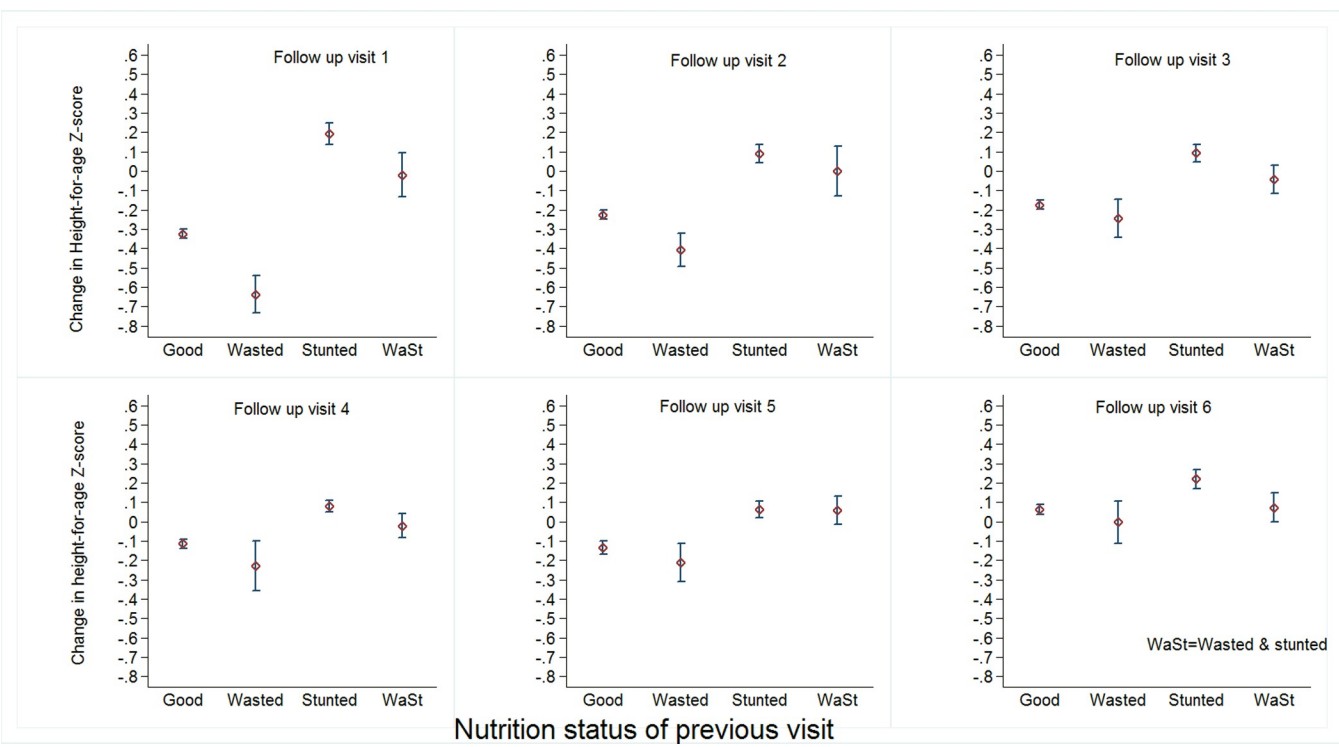

**Fig 1. Linear growth velocity for non-wasted/non-stunted, only wasted, only stunted and wasted and stunted children at the different follow-up visits.**

(Table 3). Also, having a MUAC ≥of 12.5 cm was associated with a positive change in HAZ, while non-wasted children had, on average, a decrease in HAZ (Table 3).

The analysis per follow up visit (repeated cross-sectional analysis), allowing the introduction of the previous wave trend in WHZ in the multivariate model, showed a delayed positive association between WHZ change and HAZ change. The effect size of the WHZ change during the previous follow up period on change in HAZ in the subsequent follow period varied across the FVs from +0.10 to +0.22 HAZ units, consistent with a positive relationship between linear growth and an increase in WHZ (S1 Table). Similarly, the change in MUAC in the preceding 4 months was positively associated with linear growth across all the FVs, with the effect size varying between +0.04 and +0.14 HAZ units (S1 Table).

**Table 3. Factors associated with height-for-age Z-score change overtime (N = 14226 and number of included children = 4629).**

| Variables | Univariate analysis[1] | | | Multivariate analysis[1] | | |
|---|---|---|---|---|---|---|
| | Coef[2] | (95%CI[3]) | p | Coef[2] | (95%CI[3]) | p-value |
| **Baseline variables** | | | | | | |
| **Sex of the child** | | | | | | |
| Boys/Girls | 0.026 | (0.007; 0.046) | 0.008 | 0.019 | (0.01; 0.038) | 0.041 |
| **Age at VO (months)** | | | | | | |
| 0-5/12-23 | -0.204 | (-0.226; -0.181) | <0.001 | -0.237 | (-0.261; -0.214) | <0.001 |
| 6-11/12-23 | -0.112 | (-0.137; -0.087) | <0.001 | -0.091 | (-0.114; -0.067) | <0.001 |
| **Mother formal education** | | | | | | |
| 7–9 years/0-6 years | -0.006 | (-0.031; 0.018) | 0.618 | 0.010 | (-0.012; 0.033) | 0.378 |
| ≥10 years/0-6 years | -0.001 | (-0.032; 0.031) | 0.963 | 0.034 | (0.005; 0.064) | 0.0023 |
| **Toilet type at V0** | | | | | | |
| Improved /Unimproved | -0.064 | (-0.085; -0.082) | <0.001 | -0.062 | (-0.084; -0.040) | <0.001 |
| **Water source type at V0** | | | | | | |
| Improved/ Unimproved | -0.048 | (-0.072; -0.024) | <0.001 | -0.031 | (-0.055; -0.008) | <0.001 |
| **MUAC[6] category at V0** | | | | | | |
| MUAC≥12.5 cm/MUAC<12.5 cm | 0.031 | (0.007; 0.055) | 0.012 | 0.008 | (-0.018; 0.035) | 0.539 |
| **Wasted at V0** | | | | | | |
| No WHZ[7] ≥-2/Yes: WHZ[7] <-2 | -0.010 | (-0.039; 0.019) | 0.500 | -0.063 | (-0.094; -0.033) | <0.001 |
| **Stunted at V0** | | | | | | |
| No HAZ[8] ≥-2/Yes: HAZ[7] <-2 | -0.230 | (-0.250; -0.211) | <0.001 | -0.167 | (-0.187; -0.148) | <0.001 |
| **Time varying variables** | | | | | | |
| MUAC change between visits (cm) | 0.073 | (0.061; 0.085) | <0.001 | 0.162 | (0.150; 0.174) | <0.001 |
| WHZ change between visits | -0.189 | (-0.203; -0.176) | <0.001 | -0.279 | (-0.293; -0.266) | <0.001 |
| **MUAC group** | | | | | | |
| MUAC≥12.5 cm/MUAC<12.5 cm | 0.132 | (0.089; 0.175) | <0.001 | 0.125 | (0.081; 0.169) | <0.001 |
| **Wasted** | | | | | | |
| No Z-score ≥-2/Yes: Z-score <-2 | -0.046 | (-0.080; -0.013) | 0.006 | 0.038 | (0.003; 0.074) | 0.034 |

[1]Computed using multilevel regression analysis test (stata mixed command);

[2]Coef = coefficient;

[3]CI = confidence Interval;

[4]HAZ = Length/height-for-age Z-score (2006 WHO reference curves); stunted if HAZ<-2;

[5]V0 = recruitment into the cohort visit;

[6]MUAC = Mid-Upper Arm Circumference [7]WHZ = weight-for-length/height Z-score (2006 WHO reference curves); wasted = yes if WHZ<-2; [7]HAZ = Height/length-for-Age Z-score (2006 WHO reference curves); stunted = yes if WHZ<-2.

## Association between wasting parameters (presence of wasting, WHZ change, MUAC change) and accelerated linear growth

A total of 2965 episodes of accelerated linear growth, defined by an increase of $\geq 0.67$ between two follow up visits, were observed during the study period representing 17.2% (2965/17212) of the measurements. This prevalence varied from 1.98% to 6.63% across FVs, with the highest observed at V0 and the lowest at FV4 (S2 Table). Close to half of the children had at least one episode of accelerated linear growth [95% CI = 47.4 (46.1; 48.8) %]. The proportion of children who experienced at least one episode of accelerated linear growth was higher among children already stunted at V0 (54.2% [51.0; 57.4%] versus 45.6% [43.8; 47.5%] for those not stunted at recruitment; $\Delta$ (95%CI) = 8.6 (5.2–12.0) %; $p<0.001$)].

The univariate and multivariate associations between accelerated linear growth and the presence of wasting as measured by WHZ and MUAC, presence of stunting at V0, change in WHZ, and change in MUAC are shown in Table 4. A cm increase in MUAC between follow up visits more than doubled the likelihood of occurrence of accelerated linear growth, while an increase of 1 unit of WHZ more than halved this likelihood (Table 4). Overall, MUAC group and wasting status at FVs were not independently associated with the occurrence of accelerated linear growth (Table 4). Non-wasted children at baseline were less likely to experience an accelerated linear growth than those who were wasted (Table 4). Similarly, children not-stunted enrolment were less likely to have accelerated linear growth during the study period (Table 4). The cross-sectional analyses allowing the inclusion of ante-previous status showed that having a WHZ <-2 at the *ante*-previous follow-up visit was consistently associated with a reduced likelihood of experiencing an accelerated linear growth (S2 Table). For WHZ change after FV1, an increase of WHZ by one unit during the *ante*-previous was associated with a statistically significant two to three-fold increase in the likelihood of experiencing an episode of accelerated linear growth (S2 Table).

## Discussion

### Summary of findings

Firstly, the findings of this study confirm the high prevalence of stunting among Cambodian infants and young children. The prevalence was high at recruitment into the cohort and increased over time in both sexes. Secondly, our findings show a strong association between the presence of wasting, WHZ change and MUAC change on subsequent linear growth differences and the likelihood of experiencing accelerated linear growth. This finding shows that wasting parameters predicted linear growth.

### Interpretation of the findings

Studies conducted in Africa have recently demonstrated a relationship between an episode of wasting and the occurrence of stunting a few months later [18, 25]. To the best of our knowledge, there has not been any study demonstrating an interrelationship between change in wasting parameters and linear growth during childhood. Our study is the first to demonstrate that correction of wasting may induce catch up linear growth a few months later. This conclusion is strengthened by the observed strong positive association between HAZ gain and the indicator of weight replenishment used in our analysis, WHZ and MUAC gain, which has not previously been reported among children followed up prospectively.

Despite experiencing, a slow decline in the prevalence of undernutrition in all its forms, Cambodia [42] is still classified as one of the countries with a very high level of stunting in the region [43, 44]. Our study showed an unchanged prevalence of stunting for the same age groups

**Table 4. Association between stunting, wasting parameters and occurrence of accelerated linear growth in the 4630 children included in the analysis (n = 14340 & n episodes = 2965).**

| Variables | Univariate analysis[1] | | | Multivariate analysis[1] | | |
|---|---|---|---|---|---|---|
| | OR[2] | (95%CI[3]) | p | OR[2] | (95%CI[3]) | p-value |
| **Baseline variables** | | | | | | |
| **Sex of the child** | | | | | | |
| Boys/Girls | 1.22 | (1.03; 1.45) | 0.018 | 1.21 | (1.08; 1.35) | 0.001 |
| **Age at VO (months)** | | | | | | |
| 0-5/12-23 | 1.97 | (1.67; 2.32) | <0.001 | 1.29 | (1.10; 1.52) | 0.001 |
| 6-11/12-23 | 1.10 | (0.83; 1.45) | 0.518 | 1.12 | (0.76; 1.64) | 0.573 |
| **Mother formal education** | | | | | | |
| 7–9 years/0-6 years | 0.97 | (0.79; 1.18) | 0.736 | 1.03 | (0.82; 1.30) | 0.786 |
| ≥10 years/0-6 years | 0.97 | (0.78; 1.21) | 0.807 | 1.21 | (0.92; 1.59) | 0.180 |
| **Toilet type at V0** | | | | | | |
| Improved /Unimproved | 1.30 | (1.01; 1.68) | 0.045 | 1.10 | (0.83; 1.45) | 0.494 |
| **Water source type at V0** | | | | | | |
| Improved/ Unimproved | 1.21 | (0.81; 1.82) | 0.350 | 1.20 | (0.64; 2.23) | 0.570 |
| **MUAC[6] group at V0** | | | | | | |
| MUAC≥12.5 cm/MUAC<12.5 cm | 1.73 | (1.46; 2.06) | <0.001 | 1.10 | (0.84; 1.45) | 0.471 |
| **Wasted at V0** | | | | | | |
| No WHZ[7] ≥-2/Yes: WHZ[7] <-2 | 0.73 | (0.64; 0.84) | <0.001 | 0.91 | 0.82; 1.00) | 0.058 |
| **Stunted at V0** | | | | | | |
| No HAZ[8] ≥-2/Yes: HAZ[7] <-2 | 0.52 | (0.36; 0.74) | <0.001 | 0.54 | (0.37; 0.80) | 0.002 |
| **Time varying variables** | | | | | | |
| MUAC change between visits (cm) | 1.80 | (1.59; 2.04) | <0.001 | 2.15 | (2.07; 2.22) | <0.001 |
| WHZ change between visits | 0.44 | (0.40; 0.47) | <0.001 | 0.40 | (0.37; 0.44) | <0.001 |
| **MUAC group** | | | | | | |
| MUAC≥12.5 cm/MUAC<12.5 cm | 0.89 | (0.56; 1.39) | 0.603 | 0.72 | (0.38; 1.36) | 0.313 |
| **Wasted** | | | | | | |
| No Z-score ≥-2/Yes: Z-score <-2 | 1.31 | (1.07; 1.62) | 0.010 | 1.03 | (0.91; 1.16) | 0.637 |

[1]Computed using multilevel regression analysis test (stata melogit command);

[2]OR = Odds Ratio;

[3]CI = confidence Interval;

[4]HAZ = Length/height-for-age Z-score (2006 WHO reference curves); stunted if HAZ<-2;

[5]V0 = recruitment into the cohort visit;

[6]MUAC = Mid-Upper Arm Circumference [7]WHZ = weight-for-length/height Z-score (2006 WHO reference curves); wasted = yes if WHZ<-2; [7]HAZ = Height/length-for-Age Z-score (2006 WHO reference curves); stunted = yes if WHZ<-2.

since the 2014 Cambodia Demographic Health Survey (CDHS) [11, 45], confirming the results of a study that compared the rates found in successive CDHS [44]. This stunting decline rate of only around 1.1% per year only from 2010 to 2016 (39,2% to 32,5%) is insufficient for Cambodia to meet the World Health Assembly target of reducing the number of stunted by 40% by 2025 [1, 46]. Thus a review of current interventions and scaling up of interventions that can have an impact on accelerating the reduction of stunting are urgently required. A collaboration of private and public health sectors in delivering health and nutrition services, adoption of age-appropriate feeding practices at scale, and targeting of the poor with safety net programs have been suggested as effective strategies for reducing stunting in South East Asia, including Cambodia, even if the impact of such interventions for Cambodia in not clear today [18].

The detrimental short and long term consequences of undernutrition, including stunting, on survival and development during infancy and early childhood, are well documented [21, 47]. In our study, the prevalence of stunting was already high to very high at recruitment regardless of the age group cohort for children under the age of 24 months. This suggests that children are stunted at the most vulnerable period of their brain development, a piece of further evidence on the need to address stunting in Cambodia urgently [21, 47]. Some of the long-term health consequences of undernutrition include an increase in the risk of chronic non-communicable diseases (NCDs) such as obesity, diabetes and hypertension. The prevalence of these NCDs is on the rise in South East Asia, including Cambodia [44, 48]. From a human capital development perspective, a reduced potential in educational attainment due to undernutrition and countries' economic growth has been demonstrated as well [49–52].

The high prevalence of stunting at recruitment for children in the age group cohort of 0–5 months suggests that prenatal factors play an essential role in its occurrence. Parallelly, the increase in the stunting prevalence over time indicates that postnatal factors also contribute to the maintenance and development of stunting. Analyses conducted using the same data used for this study by other authors have identified several postnatal determinants of stunting in our study population, including inappropriate feeding practices, increased exposure to animal faeces leading to Giardia Duodenalis infection, high incidence of infections and inadequate water and sanitation practices [26–28, 53]. Also, an interaction between wasting, infection and stunting in Cambodian under-five children has previously been documented [27].

Another study conducted in Cambodia showed that both feeding practices for mothers during pregnancy and lactation period and children below 14 months were inadequate in urban and rural settings of the country [54]. Interventions to address these determinants exist in Cambodia and are all scalable [20, 22]. Based on our findings, we advocate for their rapid scale-up to accelerate stunting reduction in Cambodia.

We observed a positive correlation between weight replenishment, measured by WHZ or MUAC change, and linear catch-up growth even though only a small proportion of wasted children in our study reported having received the recommended supplementary or therapeutic feeding treatment. It is thus likely that if all the wasted children had received the appropriate treatment, the increase in stunting prevalence over time might have been reduced. A study conducted in Burkina Faso showed that stunted children with wasting respond well to recommended treatment [55]. Increasing investment in the prevention and treatment of wasting in Cambodia will likely contribute to an accelerated reduction of stunting. Prevention and treatment of wasting should be part of the national package of critical interventions to address stunting. This recommendation is also supported by the fact that around 14 to 19% of the stunted children in this study were also wasted.

Our results support previously reported evidence suggesting that linear catch-up growth and weight catch up growth occurs with a lag of approximately three months. This suggests that the absence of effect of the treatment of wasting on linear growth reported in many earlier studies was because the height gain outcome was measured too early in the course of nutrition recovery [7, 17, 24, 56]. Our results, those of Isanaka et al., Lelijveld et al., and that of Schoenbuchner et al., who have examined this question recently, are in favour of introducing a systematic follow up after discharge at three and or 6 six months after the wasting correction criteria have been met and to collect data to measure the effect of wasting correction on catch up linear growth [17, 24, 56]. This will contribute to increasing the body of evidence on the need to link stunting and wasting programming. It will also create the possibility of including a new component in the guidelines such as a follow up multiple micronutrients supplementation to support the linear catch-up growth, the immune system recovery, address persisting iron deficiency and other micronutrient deficiencies [57, 58]. Such follow up is already

advocated for by the report of high relapse rate and mortality during the first three to six months following discharge from wasting correction programmes [59, 60].

Although both MUAC changes and WHZ changes are indicators of tissue deficit replenishment, in our study, we observed a difference in the time lags when both MUAC and WHZ change correlate to the same height change, with MUAC change having the shortest time lag. This finding has not been reported before and is thus difficult to interpret given that both MUAC and WHZ contribute to wasting correction and increase in body mass [61]. A plausible explanation is that this discrepancy is just a reflection of the weak correlation between WHZ change and MUAC change observed in our study, indicating that the two parameters selected different children (S1 Fig). A similar discrepancy between MUAC change and weight change in children recovering from wasting has been reported in the literature [62]. However, other authors have also reported a good correlation between MUAC and weight changes during recovery from severe wasting [62–64]. Nonetheless, this finding has a crucial programmatic implication. It indicates that MUAC change may be an early indicator of response to the stunting reduction interventions, although both MUAC change and weight change should always be used.

Currently, stunting is viewed as a chronic form of undernutrition that, once established, is largely irreversible, especially after two years of age, and that can only be addressed in developmental context by long-term preventive interventions [65, 66]. Our analysis showed that stunted children had accelerated linear growth suggests that many of the stunted children retained the potential to catch up on growth, with some achieving complete recovery. This finding calls into question the general thinking that stunting can be addressed only by preventive interventions. As demonstrated by our findings, interventions to support the potential of recovery from stunting are also critical. Studies that showed that recovered stunted children had similar cognitive capacity to those who never experienced stunting further support the need to add curative approaches to the package of interventions to reduce stunting [47, 67, 68].

## Strengths and limitations

The results described in this paper need to be interpreted, taking into account the strengths and weaknesses of the data. The study's main strength is the longitudinal design that allowed follow-up of the same children prospectively and calculation of the estimates of interests for each follow-up visit. A second important strength is that the study was conducted in the three different livelihood contexts of Cambodia and in a "real world" environment; thus, the results can be generalized to the Cambodian context and other similar settings. The third strength is the cohort starting sample of over 5000 children, and a median number of data collection points per child of 5 out of the seven planned provided enough power for calculating key outcomes at all the follow-up visits.

However, this was a longitudinal open cohort study with many of those included in the analysis enrolled either at V0 or at FV5. This may have introduced a seasonal bias that we could not assess in the present study. It was also not a birth cohort; thus, the contribution of prenatal growth factors might be overstated. Also, we could not attach the growth pattern of a period to a particular health event or intervention. This hindered the adjustment of the observed associations between the wasting parameters and the linear growth outcomes. However, this is common in many community-based studies. Also, previous publications from the original analysis of these data addressed some of these aspects [27, 28, 53, 69]. These papers have suggested that infections may be contributing to the occurrence of stunting [27, 53, 69]. However, in our analysis for children below two years at enrolment, type of toilet and source of drinking water were not consistently associated with the linear growth parameters studied

departing from Manzoni et al. conclusion [53]. Finally, we could not adjust for wealth and household food security as these variables were not available. Despite these limitations, we consider that the results of our analyses and the derived conclusions are plausible, given that they are in line with the most recently published observations. The strength of the associations are high, and the adjustment with the variables available did not reduce the strength of the observed associations.

## Conclusion

We provide evidence of a positive relationship between wasting parameters and linear growth. Our findings have underscored several points of programming and policy relevance, including 1) wasting was a contributing factor to stunting and prevention of wasting may contribute to reducing stunting in Cambodia; 2) WHZ and MUAC increases were positively correlated with accelerated linear growth suggesting wasting correction can contribute to preventing and reversing stunting; 3) Linear catch-up growth can be observed 3 to 4 months from the time of wasting correction, suggesting a need to include a post-wasting recovery follow up period in the guidelines for the management of wasting. Our findings support the few studies recommending a re-thinking of the current approach of wasting and stunting separation in terms of policy, programming, and financing and a move towards a more integrated approach to maximize the effectiveness of the interventions targeting both wasted and stunted children [17, 24, 65]. This will help accumulate evidence of the effect of SAM wasting treatment on linear catch-up growth, serve as a period for promoting catch up linear growth and corrections of persisting deficiencies.

The evidence in our study strongly supports the integration of both preventative and curative wasting and stunting programming, given the multiplicative effect they have on the prevalence and long-term effects of undernutrition and overnutrition and other metabolic syndrome-related diseases. Such integration has the potential of accelerating the attainment of World Health Assembly targets on undernutrition in a country where both wasting and stunting are of public importance, such as in Cambodia [6]. Such studies should also be repeated in other contexts such as South-East Asia.

## Supporting information

**S1 Fig. Overlap between weight-for-height Z-score (WHZ) and Mid-Upper Arm Circumference (MUAC) change categories during the different periods.** WHZ change and MUAC change at visit 0 to follow up visit 6.
(PDF)

**S1 Table. Factors associated with height-for-age Z-score (HAZ) change at the different follow-up visits.** HAZ change from visit 0 to follow up visit 6.
(DOCX)

**S2 Table. Association between stunting, wasting parameters and occurrence of accelerated linear growth at the different follow up visits.** Change in wasting and stunting parameters from visit 0 to follow up visit 6.
(DOCX)

**S1 File.**
(XLS)

**S2 File.**
(XLS)

**S3 File.**
(XLSX)

**S4 File.**
(CSV)

**S5 File.**
(XLSX)

**S6 File.**
(CSV)

**S7 File.**
(CSV)

## Acknowledgments

The authors would like to thank Roland Kupka for his insightful comments that improved the manuscript. We also thank Selamawit Negash and Sophonneary Prak for facilitating access permissions to the data sets. Thanks also to Gabriela Hondru for her support in extracting variables for this study from the MyHealth study database.

**Disclaimer:** Mueni Mutunga is a UNICEF staff member. The opinions and statements in this article are those of the author and may not reflect official UNICEF policies.

## Author Contributions

**Conceptualization:** Mueni Mutunga, Alexandra Rutishauser-Perera, Sophonneary Prak, Paluku Bahwere.

**Data curation:** Paluku Bahwere.

**Formal analysis:** Paluku Bahwere.

**Project administration:** Alexandra Rutishauser-Perera.

**Supervision:** Mueni Mutunga.

**Validation:** Mueni Mutunga, Frank T. Wieringa.

**Visualization:** Mueni Mutunga, Paluku Bahwere.

**Writing – original draft:** Mueni Mutunga, Paluku Bahwere.

**Writing – review & editing:** Mueni Mutunga, Alexandra Rutishauser-Perera, Arnaud Laillou, Jacques Berger, Frank T. Wieringa, Paluku Bahwere.

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
