## [Decision Letter · Decision Letter 0]

8 Mar 2021

PONE-D-20-36251

The relationship between wasting and stunting in Cambodian children: Secondary data analysis of longitudinal data of children below 24 months of age followed up until the age of 59 months.

PLOS ONE

Dear Dr. Mutunga,

Thank you for submitting your manuscript to PLOS ONE. After careful consideration, we feel that it has merit but does not fully meet PLOS ONE’s publication criteria as it currently stands. Therefore, we invite you to submit a revised version of the manuscript that addresses the points raised during the review process.

We look forward to receiving your revised manuscript.

Kind regards,

Srinivas Goli, Ph.D.

Academic Editor

PLOS ONE

Journal Requirements:

3. Please provide the full name of the relevant authority that granted authorization to use the MyHealth data.

Additional Editor Comments:

Considering favourable opinions from the reviewers, I am going with a decision of minor revision for this paper. Looking forward to revised version.

Reviewers' comments:

Reviewer's Responses to Questions

**Comments to the Author**

1. Is the manuscript technically sound, and do the data support the conclusions?

Reviewer #1: Partly

Reviewer #2: Yes

Reviewer #3: Yes

2. Has the statistical analysis been performed appropriately and rigorously? 

Reviewer #1: No

Reviewer #2: Yes

Reviewer #3: Yes

3. Have the authors made all data underlying the findings in their manuscript fully available?

Reviewer #1: Yes

Reviewer #2: No

Reviewer #3: Yes

4. Is the manuscript presented in an intelligible fashion and written in standard English?

Reviewer #1: Yes

Reviewer #2: Yes

Reviewer #3: Yes

5. Review Comments to the Author

Reviewer #1: The authors conducted an interesting study dealing with an important demographic and epidemiological topic. However, all in all the paper is very (too) lengthy and has a lot of repetitions. The paper raised too many objectives, which makes it very confusing to read particularly in the result section. The paper would benefit from focusing on one of the objectives only, say on finding important determinants explaining the relationship between wasting and stunting without taking additionally the trends over time into account. Moreover, there are several major concerns that need to be tackled first, before going to minor revisions. Especially, the used statistical methods, the result reporting, the discussion and the structure of the paper need to be revised fundamentally in my opinion:

Major concerns:

1. Paper structure and focus/ objective

Please focus on one specific research question/ aim in your study. The paper is very lengthy, also your tables and result reporting in general, and is thus very confusing to read. Please avoid to show lengthy tables and try to present your results in a clearer way (e.g. use more figures). Try to report the main results only and move the other results to the appendix.

2. Variable selection

The entire manuscript makes the impression that the authors performed a prediction model and not an explanation model, and this is the case due the authors’ variable selection remained unclear. The authors write that variable selection was based on the given/ used dataset (MyHealth). This sounds like they applied a data-driven approach to choose the relevant third variables. However, in fact the authors did not. Please, justify your variable selection by citing previous studies/ literature on wasting and stunting, and not by using variables which “just available” in your data.

3. Methods/ Statistical analysis

This part is rather short and needs more details on the used statistical approach and the underlying study design, which was elaborated based on the given data structure.

- The authors write that they performed mixed models to control for the fact that the data are of hierarchical nature, which is per se correct. However, the authors write in previous parts of the paper that they used several observations per individual, which means that panel data were used. Unfortunately, the authors seem to not using appropriate methods to control for clustering over time (panel model). Not regarding this leads biased standard errors. I strongly recommend to perform suitable panel analysis models, which means in your case to add a random effect to the model controlling for intra-individual clustering over time.

- The authors write that they used mixed models. Please clarify which kind of mixed models you have calculated. For me, it seems that continuous and dichotomous outcome variables were used. If so, you probably estimated logistic and linear models…

- Please add formulas to this part explaining your statistical approach. You explained that multi-level-modeling was applied – which levels were included and used as random effects in the models?

4. Study design

The study design and particularly the definition of your baseline assessment does not seem to be correct or say, your data handling is problematic with view to that. You used unbalanced panel data from MyHealth, which means that there are varying baseline assessments for the participants. Say one has baseline assessment in wave 1, one has it in wave 2, and another has the baseline assessment in wave 3. Your results, especially those shown Figure 2, evoke the impression that baseline was for you generally the first wave which was conducted in MyHealth. This is wrong since the baseline is not a fix wave or point in time when using unbalanced panel data, but a variable/ time-varying point in time. Please explain, and if necessary, change your data with view to this.

5. Discussion

The discussion is poorly written and structured. Please, give the discussion a clear structure with sub-parts, e.g.:

a) Summary of the findings

b) Interpretation of the findings

c) Strengths and limitations

d) Conclusion

- Additionally, your reflection of the own study results (limitation part) is too positively accentuated. Your discussion makes the impression you have found real causal effects, which is with view to your used statistical methods, not true. Your approach is barely appropriate to find out real causality when comparing to other approaches (fixed effects modeling, g-methods etc.). For example, you write “our results confirm that” – please avoid such phrasing. Better is to write “Our results suggest that…”.

Reviewer #2: The efforts of the authors are commended. However, there are a number of issues to be addressed to improve the quality of the manuscript to make it fit for publication in PLOS ONE journal.

Abstract:

The objectives are not clearly stated in the abstract section. It appears it is only the first objective that was stated in the abstract. The last three lines under the introduction clearly spelt out the objectives. I think this can be moved to the abstract as well, while the research questions stated under methods can be moved up to the ‘introduction’ section.

Introduction:

The authors can provide a brief background/ information on the three provinces where the data collection took place rather than mentioning it passively under the “Methods” section. The authors can as well elaborate on the findings/weaknesses of the previous studies that informed the present study.

Methods:

There is repetition about the period of data collection. Check the second line and the last line of the ‘Study design and participants’ under the ‘Methods’ section.

I feel the key research questions stated under ‘Outcomes of interest and sample size’ under the ‘Methods’ section ought to be moved up to the ‘Introduction’ section.

Results:

Since the authors stated 1992 children were excluded in the data analysis, I see no reason why they should disintegrate the number again under the ‘Study participants’ in the Results section. More so, that the exclusion criteria have been clearly stated in the ‘Study design and participants’ under the ‘Methods’ section. Similarly, presenting results for the number of children excluded makes the result on Table 1 very clumsy and confusing. I feel the authors can exclude results for the excluded children. This should also be removed from the table under the results for ‘children’ on page 11 (≥24months – 0/5172).

Discussion:

The discussion is too long! The findings of the study should be discussed viz-a-viz the stated objectives with regards to the reviewed literature, highlighting the strengths and weaknesses (if any) of the study.

Other comments:

There should be consistency with the use of terms, especially the terms wasting and stunting. Since these two terms have been clearly defined under the ‘introduction’, they should be used consistently throughout the body of the manuscript.

There are few grammatical errors to be corrected as well.

Reviewer #3: The introduction appropriately detailed available findings and justifies the need for this study. The data analysis shows an excellent tracking of nutritional status, figures, tables and result provided appropriate information at each follow-up visits. Using a cohort design and replicating standard measurement levels for essential variables and the adequacy of the sample size is worthy of note. The evidence provided in the discussion section are well substantiated in the data analysis. This research strengthens the need for joint program action to address the interrelationship between wasting and stunting.

It will be clearer to readers if the sample distribution over the phases is presented in a table, it will show the sample variation over the study period and how children and enter and exit. This can also guide replicability of the study.

6. PLOS authors have the option to publish the peer review history of their article (what does this mean?). If published, this will include your full peer review and any attached files.

Reviewer #1: No

Reviewer #2: No

Reviewer #3: No

---

## [Author Response · Author response to Decision Letter 0]

17 May 2021

Reviewer 1

"Paper structure and focus/ objective

Please focus on one specific research question/ aim in your study. The paper is very lengthy, your tables and result reporting in general, and is thus very confusing to read. Please avoid showing lengthy tables and trying to present your results in a clearer way (e.g. use more figures). Try to report the main results only and move the other results to the appendix."

We thank the reviewer for the comment. We, however, would like to explain that our view was to avoid the dispersion of the findings by disseminating them via several papers. Giving a comprehensive understanding and full picture of the study and results in one paper was our objective. We were also convinced that the length of the paper and tables was acceptable to PLOS ONE journal. Finally, with the rising interest in systematic review and meta-analysis, we believe that giving the results in detailed tables rather than in pictorial form facilitates the further use of reported information. 

"Variable selection

The entire manuscript makes the impression that the authors performed a prediction model and not an explanation model, and this is the case due the authors' variable selection remained unclear. The authors write that variable selection was based on the given/ used dataset (MyHealth). This sounds like they applied a data-driven approach to choose the relevant third variables. However, in fact, the authors did not. Please, justify your variable selection by citing previous studies/ literature on wasting and stunting, and not by using variables which "just available" in your data."

Thank you for this comment. Because this study is a secondary analysis, we have amended the text (line 182)

"Methods/ Statistical analysis

This part is rather short and needs more details on the used statistical approach and the underlying study design, which was elaborated based on the given data structure.

- The authors write that they performed mixed models to control for the fact that the data are of hierarchical nature, which is per se correct. However, the authors write in previous parts of the paper that they used several observations per individual, which means that panel data were used. Unfortunately, the authors seem to not using appropriate methods to control for clustering over time (panel model). Not regarding this leads biased standard errors. I strongly recommend to perform suitable panel analysis models, which means in your case to add a random effect to the model controlling for intra-individual clustering over time.

We did not perform panel data analysis. We reported the results of analyses using 2 data points only at each follow-up. The use of panel data analysis was made difficult by the cross-sectional/open cohort type of the data collection applied in the original study. Also, the level of missings when trying to apply panel data analysis suggested that the obtained estimates would likely be biased.

"

- The authors write that they used mixed models. Please clarify which kind of mixed models you have calculated. For me, it seems that continuous and dichotomous outcome variables were used. If so, you probably estimated logistic and linear models…"

We did specify this in the revised version in the data management and analysis part (from line 181)

"- Please add formulas to this part explaining your statistical approach. You explained that multi-level-modeling was applied – which levels were included and used as random effects in the models?"

We thank the reviewer for this comment. While we agree that the description of statistical tests is succinct, we believe that the provided information is sufficient for the targeted audience. However, we have amended the paragraph on statistical analysis and provided the stata command used and the relevant references so that those needing more details on the tests used for fitting the models can get the information from the STATA manual.

"Study design

The study design and particularly the definition of your baseline assessment does not seem to be correct or say, your data handling is problematic with view to that. You used unbalanced panel data from MyHealth, which means that there are varying baseline assessments for the participants. Say one has baseline assessment in wave 1, one has it in wave 2, and another has the baseline assessment in wave 3. Your results, especially those shown Figure 2, evoke the impression that baseline was for you generally the first wave which was conducted in MyHealth. This is wrong since the baseline is not a fix wave or point in time when using unbalanced panel data, but a variable/ time-varying point in time. Please explain, and if necessary, change your data with view to this."

Thank you for this comment. We think that this comment is related to that of the data analysis approach addressed above. As mentioned in the text and above, we re-organised the data and as correctly picked by the reviewer, the recruitment visit was the visit 0 (baseline/recruitment) for all children. We did not opt for panel data analysis for the reasons mentioned above.

"Discussion

The discussion is poorly written and structured. Please, give the discussion a clear structure with sub-parts, e.g.:

a) Summary of the findings

b) Interpretation of the findings

c) Strengths and limitations

d) Conclusion.

We thank the reviewer for this comment, but we think it is a matter of personal preference as the structure seems to have been well understood by the reviewer and the other reviewers. Also, we have noted that the journal recommends avoiding the multiplying level of sub-heading.

- Additionally, your reflection of the own study results (limitation part) is too positively accentuated. Your discussion makes the impression you have found real causal effects, which is with view to your used statistical methods, not true. Your approach is barely appropriate to find out real causality when comparing to other approaches (fixed effects modeling, g-methods etc.). For example, you write "our results confirm that" – please avoid such phrasing. Better is to write "Our results suggest that…".

We have implemented this recommendation by clarifying more what we meant by "confirm".

Reviewer 2

Abstract:

The objectives are not clearly stated in the abstract section. It appears it is only the first objective that was stated in the abstract. The last three lines under the introduction clearly spelt out the objectives. I think this can be moved to the abstract as well, while the research questions stated under methods can be moved up to the 'introduction' section.

We thank the reviewer for this suggestion. We have made the recommended change in the abstract section. For the second point, our preference is to keep the research questions in the methods section.

"Introduction:

The authors can provide a brief background/ information on the three provinces where the data collection took place rather than mentioning it passively under the "Methods" section. The authors can as well elaborate on the findings/weaknesses of the previous studies that informed the present study."

Thank you for this recommendation. The text has been amended accordingly but kept in the method section (from line 72). We believe that the reviewer recommendation is addressed in the second paragraph of the introduction for the second point. This paragraph has been written taking the length of the paper into account.

"Methods:

There is repetition about the period of data collection. Check the second line and the last line of the 'Study design and participants' under the 'Methods' section.

I feel the key research questions stated under 'Outcomes of interest and sample size' under the 'Methods' section ought to be moved up to the 'Introduction' section."

We thank the reviewer for spotting this repetition. We have deleted the last phrase of the section study design and participants. For the second point, thank you for the suggestion, but our preference is to keep this information in the methods section.

"Results:

Since the authors stated 1992 children were excluded in the data analysis, I see no reason why they should disintegrate the number again under the 'Study participants' in the Results section. More so, that the exclusion criteria have been clearly stated in the 'Study design and participants' under the 'Methods’ section. Similarly, presenting results for the number of children excluded makes the result on Table 1 very clumsy and confusing. I feel the authors can exclude results for the excluded children. This should also be removed from the table under the results for ‘children’ on page 11 (≥24months – 0/5172).”

We thank the reviewer for this comment. We have implemented your recommendation and adapted table 1.

“Discussion:

The discussion is too long! The findings of the study should be discussed viz-a-viz the stated objectives with regards to the reviewed literature, highlighting the strengths and weaknesses (if any) of the study.”

We thank the reviewer for this comment. Our opinion is that the discussion already includes the reviewer's suggestions. As indicated in the introduction, there is only one study that had previously covered the same topic. Additionally, we deliberately restricted the discussion to key points to limit the length of the paper.

Other comments:

There should be consistency with the use of terms, especially the terms wasting and stunting. Since these two terms have been clearly defined under the ‘introduction’, they should be used consistently throughout the body of the manuscript.

There are few grammatical errors to be corrected as well.

Thank you, we have corrected the grammatical errors. Regarding the use of wasting and stunting, we couldn’t identify the inconsistencies apart from the use of stunting vs stunted in appropriate locations.

Reviewer 3

The introduction appropriately detailed available findings and justifies the need for this study. The data analysis shows an excellent tracking of nutritional status, figures, tables and result provided appropriate information at each follow-up visits. Using a cohort design and replicating standard measurement levels for essential variables and the adequacy of the sample size is worthy of note. The evidence provided in the discussion section are well substantiated in the data analysis. This research strengthens the need for joint program action to address the interrelationship between wasting and stunting.

Thank you for those positive comments.

It will be clearer to readers if the sample distribution over the phases is presented in a table, it will show the sample variation over the study period and how children and enter and exit. This can also guide replicability of the study.

We agree with the reviewer that the proposed table can improve readability but given the recommendation to limit the length of the paper by the other reviewers, we have decided against adding the suggested table.

---

## [Decision Letter · Decision Letter 1]

2 Jul 2021

PONE-D-20-36251R1

The relationship between wasting and stunting in Cambodian children: Secondary data analysis of longitudinal data of children below 24 months of age followed up until the age of 59 months.

PLOS ONE

Dear Dr. Mutunga,

Thank you for submitting your manuscript to PLOS ONE. After careful consideration, we feel that it has merit but does not fully meet PLOS ONE’s publication criteria as it currently stands. Therefore, we invite you to submit a revised version of the manuscript that addresses the points raised during the review process.

ACADEMIC EDITOR: One of the reviewer still has major concerns with your statistical analyses and writing style. If you can incorporate or respond to the reviewer, we can re-consider this paper. 

We look forward to receiving your revised manuscript.

Kind regards,

Srinivas Goli, Ph.D.

Academic Editor

PLOS ONE

Additional Editor Comments (if provided):

One of the reviewer still has major concerns with your statistical analyses and writing style. If you can incorporate or respond to the reviewer, we can re-consider this paper.

Reviewers' comments:

Reviewer's Responses to Questions

**Comments to the Author**

1. If the authors have adequately addressed your comments raised in a previous round of review and you feel that this manuscript is now acceptable for publication, you may indicate that here to bypass the “Comments to the Author” section, enter your conflict of interest statement in the “Confidential to Editor” section, and submit your "Accept" recommendation.

Reviewer #1: (No Response)

Reviewer #2: All comments have been addressed

2. Is the manuscript technically sound, and do the data support the conclusions?

Reviewer #1: Partly

Reviewer #2: Yes

3. Has the statistical analysis been performed appropriately and rigorously? 

Reviewer #1: No

Reviewer #2: Yes

4. Have the authors made all data underlying the findings in their manuscript fully available?

Reviewer #1: No

Reviewer #2: Yes

5. Is the manuscript presented in an intelligible fashion and written in standard English?

Reviewer #1: Yes

Reviewer #2: Yes

6. Review Comments to the Author

Reviewer #1: Dear editor,

dear authors,

thank you for the opportunity to re-review the paper again, which deals in general with a very important and interesting topic. Unfortunately, the authors did not deal with my major concerns from the first review. Most of the very helpful reviewers’ concerns (and I mean also the concerns from the other reviewers) were rejected by authors without integrating them in the paper. This is disappointing since the remarks would had improved the paper significantly.

There are still two major concerns that were (still) fully neglected by the authors although the reviewers mentioned several times. 1) statistical methods, 2) the paper structure/writing. Due to these ignored major concerns I have to recommend to reject the paper and did not consider for a further review. Let me explain this more detailed in the following paragraph:

1) Statistical methods

The authors write in lines 84-86 that children were surveyed every 3 to 4 months, so that it makes the impression the authors used panel data. The authors answered that they indeed did not perform panel data analysis. The authors’ argumentation that “the cross-sectional/open cohort type of the data collection applied in the original study. Also, the level of missings when trying to apply panel data analysis suggested that the obtained estimates would likely be biased.” is poor or did not fit with the authors data description in the source of data section. Panel analysis can also be conducted when the data were of unbalanced nature why the argumentation is not correct from my view. Additionally, which kind of study design/ analysis strategy was used? Did the authors estimate seemingly unrelated regressions? If so, why they did not mention in the Method section? Altogether, the methodology used in the paper remained still open and I have major concerns that the statistical methods were conducted properly. Altogether, the data and method section was written poor and did not sound convincing that the authors did, in fact, perform the proper statistical methods.

1) Paper structure/writing

The paper is too lengthy, which was already mentioned in review round 1. Especially the method and discussion parts were written poor. Although the reviewers mentioned that several times, the authors did not integrate and disagreed. This is a bit unprofessional kind of responding to a review. It was already mentioned by another reviewer that the research question should be mentioned in the introduction – the authors did not regard this remark. The method section mainly focused on STATA commands and did not explain the used methods in a professional fashion by using equations or, at least, correct explanations concerning the procedure. This was mentioned in my review from round 1. Both was missed in the paper, and not integrated although it was mentioned in review 1.

I recommended in my first review, how the authors could structure their discussion to improve the readability of the paper. They neglected completely! Arguing that PLOS ONE is an open format journal cannot be used as a good argument to ignore general remarks for paper improvements…

Reviewer #2: This revision has addressed most of the issues raised in the version submitted earlier. I will like to commend the authors for this. However, I feel that the discussion is still lengthy. Nonetheless, the view of other reviewers put together will guide the final decision of the academic editor. Manuscript is strongly recommended for publication.

7. PLOS authors have the option to publish the peer review history of their article (what does this mean?). If published, this will include your full peer review and any attached files.

Reviewer #1: No

Reviewer #2: No

---

## [Author Response · Author response to Decision Letter 1]

15 Sep 2021

“Statistical methods:

The authors write in lines 84-86 that children were surveyed every 3 to 4 months so that it makes the impression the authors used panel data. The authors answered that they indeed did not perform panel data analysis. The authors' argumentation that "the cross-sectional/open cohort type of the data collection applied in the original study. Also, the level of missings when trying to apply panel data analysis suggested that the obtained estimates would likely be biased." is poor or did not fit with the authors' data description in the source of the data section. Panel analysis can also be conducted when the data were of unbalanced nature, why the argumentation is not correct from my view. Additionally, which kind of study design/ analysis strategy was used? Did the authors estimate seemingly unrelated regressions? If so, why they did not mention it in the Method section? Altogether, the methodology used in the paper remained still open, and I have major concerns that the statistical methods were conducted properly. Altogether, the data and method section was written poor and did not sound convincing that the authors did, in fact, perform the proper statistical methods.

Response: We appreciate that the reviewer has taken the time to explain this recommendation further. Despite our reservations on this recommendation given the rationale that we provided in our previous feedback (that the use of panel data analysis was made difficult by the cross-sectional/open cohort type of the data collection applied in the original study; and also that panel analysis that deleted cases with missing values using a listwise approach which suggested that the obtained estimates would likely be biased), we have opted to implement the reviewers' recommendation. We have replaced the cross-sectional approach with the panel analysis (random effects) approach (table 3 and 4 and related text). The former tables are now included as supplementary tables. We have kept some aspects of the relationship between ante-previous WHZ change, better demonstrated in the cross-sectional modelling approach, in the main text as we consider this to be important information with policy and programming implications. 

2. “ Paper structure/writing

The paper is too lengthy, which was already mentioned in review round 1. Especially the method and discussion parts were written poorly. Although the reviewers mentioned that several times, the authors did not integrate and disagreed. This is a bit unprofessional kind of responding to a review. Another reviewer already mentioned that the research question should be mentioned in the introduction – the authors did not regard this remark. The method section mainly focused on STATA commands and did not explain the used methods in a professional fashion by using equations or, at least, correct explanations concerning the procedure. This was mentioned in my review from round 1. Both were missed in the paper, and not integrated, although it was mentioned in review 1.

Response: We appreciate the detailed feedback on the reasons for disagreeing with the paper structure, content and length. However, we disagree with the judgment that not agreeing with a reviewer is acting unprofessionally. We noted that the other reviewer was generally happy with how we addressed the comments except for the length. Below is a point-by-point response to the issues in question

 1) The research questions have now been moved to the introduction, and the outcomes sub-section of the methods updated accordingly. 

2) On the use of equations, we stand by our previous response based on the audience we are targeting for whom the use of equation is not necessary for paper readability

 3) For the selection procedure, we also stand to our previous response as we prefer giving a transparent account of what was done

3. I recommended in my first review how the authors could structure their discussion to improve the readability of the paper. They neglected completely! Arguing that PLOS ONE is an open format journal. This cannot be used as a good argument to ignore general remarks for paper improvements.

Response:We thoughtfully considered this suggestion considering that the other reviewers, including the external reviewers we had used prior to submitting the manuscript, did not have an issue with the readability of the discussion. Regardless, we have now implemented this recommendation.

---

## [Decision Letter · Decision Letter 2]

27 Oct 2021

The relationship between wasting and stunting in Cambodian children: Secondary data analysis of longitudinal data of children below 24 months of age followed up until the age of 59 months.

PONE-D-20-36251R2

Dear Dr. Mutunga,

We’re pleased to inform you that your manuscript has been judged scientifically suitable for publication and will be formally accepted for publication once it meets all outstanding technical requirements.

Kind regards,

Srinivas Goli, Ph.D.

Academic Editor

PLOS ONE

Additional Editor Comments (optional):

Considering my own reading and reviewers opinion, I am recommending this paper.

Reviewers' comments:

Reviewer's Responses to Questions

**Comments to the Author**

1. If the authors have adequately addressed your comments raised in a previous round of review and you feel that this manuscript is now acceptable for publication, you may indicate that here to bypass the “Comments to the Author” section, enter your conflict of interest statement in the “Confidential to Editor” section, and submit your "Accept" recommendation.

Reviewer #2: All comments have been addressed

2. Is the manuscript technically sound, and do the data support the conclusions?

Reviewer #2: Yes

3. Has the statistical analysis been performed appropriately and rigorously? 

Reviewer #2: Yes

4. Have the authors made all data underlying the findings in their manuscript fully available?

Reviewer #2: Yes

5. Is the manuscript presented in an intelligible fashion and written in standard English?

Reviewer #2: Yes

6. Review Comments to the Author

Reviewer #2: The manuscript has been greatly improved and I will like to recommend that it be accepted for publication in PLOS ONE journal.

7. PLOS authors have the option to publish the peer review history of their article (what does this mean?). If published, this will include your full peer review and any attached files.

Reviewer #2: **Yes: **Olufunmilayo Olufunmilola Banjo (PhD)

---

## [Editor Report · Acceptance letter]

9 Nov 2021

PONE-D-20-36251R2 

The relationship between wasting and stunting in Cambodian children: Secondary analysis of longitudinal data of children below 24 months of age followed up until the age of 59 months. 

Dear Dr. Mutunga:

I'm pleased to inform you that your manuscript has been deemed suitable for publication in PLOS ONE. Congratulations! Your manuscript is now with our production department. 

Kind regards, 

on behalf of

Dr. Srinivas Goli 

Academic Editor

PLOS ONE